# O-GlcNAcylation in Renal (Patho)Physiology

**DOI:** 10.3390/ijms231911260

**Published:** 2022-09-24

**Authors:** Rodrigo P. Silva-Aguiar, Diogo B. Peruchetti, Ana Acacia S. Pinheiro, Celso Caruso-Neves, Wagner B. Dias

**Affiliations:** 1Carlos Chagas Filho Institute of Biophysics, Federal University of Rio de Janeiro, Rio de Janeiro 21941-901, Brazil; 2Rio de Janeiro Innovation Network in Nanosystems for Health-NanoSAÚDE/FAPERJ, Rio de Janeiro 21045-900, Brazil; 3National Institute of Science and Technology for Regenerative Medicine, Rio de Janeiro 21941-902, Brazil

**Keywords:** O-GlcNAcylation, kidney, renal disease, albuminuria, post-translational modification, O-GlcNAc transferase, O-GlcNAcase, O-GlcNAc

## Abstract

Kidneys maintain internal milieu homeostasis through a well-regulated manipulation of body fluid composition. This task is performed by the correlation between structure and function in the nephron. Kidney diseases are chronic conditions impacting healthcare programs globally, and despite efforts, therapeutic options for its treatment are limited. The development of chronic degenerative diseases is associated with changes in protein O-GlcNAcylation, a post-translation modification involved in the regulation of diverse cell function. O-GlcNAcylation is regulated by the enzymatic balance between O-GlcNAc transferase (OGT) and O-GlcNAcase (OGA) which add and remove GlcNAc residues on target proteins, respectively. Furthermore, the hexosamine biosynthetic pathway provides the substrate for protein O-GlcNAcylation. Beyond its physiological role, several reports indicate the participation of protein O-GlcNAcylation in cardiovascular, neurodegenerative, and metabolic diseases. In this review, we discuss the impact of protein O-GlcNAcylation on physiological renal function, disease conditions, and possible future directions in the field.

## 1. Introduction

The kidneys are essential organs for the maintenance of internal compartment homeostasis [1]. Through intricate correlation between histologic, morphologic, biochemical, and anatomic features, the nephron participates in the regulation of blood pH, plasma osmolarity, and blood pressure at narrow ranges through regulated manipulation of H^+^ and HCO_3_^−^, water, and Na^+^ [1]. Furthermore, the renal tissue also contributes to energy balance through the reabsorption of glucose [2], amino acids [3], and proteins [4]; it is a major site of gluconeogenesis during starvation [5]; promotes hematopoietic differentiation and oxygen adaptation through erythropoietin production and secretion [6]. Importantly, there is an extensive crosstalk with different organs, such as the central nervous system [7], the gastrointestinal tract [8], and the cardiovascular system [9] that also modulates renal function. These tasks are performed by the exquisite regulation of filtration at the glomerulus and selective reabsorption or secretion of different solutes across the tubular segments.

Kidney diseases caused 1.2 million deaths in 2017, making it the 9th cause of death in that year [10]. Furthermore, approximately 700 million people have been diagnosed with all-stage renal diseases, with an overall prevalence of 9% [10]. Renal diseases impose a severe social and economic burden in several healthcare programs worldwide [11,12]. Thus, investigation of the molecular mechanisms involved in the genesis and progression of renal diseases is urgent for faster diagnosis and better treatment of these patients.

The internal compartment is constantly being challenged, requiring rapid responses for maintenance of the homeostasis. Renal function is highly adaptable, changing capacity rate as required [13,14,15]. From a molecular perspective, this task has evolutionarily relied on protein post-translational modifications (PTMs) [16,17]. PTM diversity grants selectivity, intensity, and temporal regulation of initiation and termination of intracellular responses to extracellular stimulus. Several PTMs have been widely studied, including phosphorylation [18], acetylation [19], and ubiquitylation [20]. In the early 1980s, an intracellular signal transduction mechanism that involves the O-linked addition of β-N-acetylglucosamine moieties (GlcNAc) on serine or threonine residues of nucleocytoplasmic proteins, so called O-GlcNAcylation, was reported [21]. Despite several reviews that have addressed the role of O-GlcNAcylation in the physiology and pathophysiology of different tissues, such as the cardiovascular [22,23], the central nervous system [24,25], and the immune system [26,27], the role of O-GlcNAcylation in the kidneys has yet not been thoroughly discussed. Here, we highlight findings describing the impact of changes in O-GlcNAcylation for renal physiology and pathophysiology and discuss future directions in the field.

## 2. O-GlcNAcylation

Initially, it was believed that covalent modifications of proteins with carbohydrates would occur only at the endoplasmic reticulum and the Golgi apparatus in proteins that are secreted or expressed at the plasma membrane in a process called N-glycosylation [28,29]. This paradigm was changed since the first description of intracellular O-GlcNAcylation outside of these compartments [21]. The molecular mechanisms governing O-GlcNAc homeostasis and its participation in physiology and in pathophysiology have been extensively studied and validated [22,23,24,25,26,27,30,31]. Two enzymes were described to be responsible for O-GlcNAc cycling: (1) the O-GlcNAc transferase, OGT, encoded by the gene OGT, responsible for adding O-GlcNAc moiety into serine or threonine residues of target proteins; and (2) the glycosidase O-GlcNAcase, OGA, encoded by the gene OGA, responsible for removing O-GlcNAc residues from proteins [30]. Both OGT and OGA are evolutionarily conserved [32], ubiquitously expressed in mammalian cells, and their structures have been resolved [33,34]. Deletion of both OGT and OGA is lethal in different animal models, suggesting that O-GlcNAcylation is essential for organism development [35,36,37].

In addition to OGA and OGT balance, O-GlcNAcylation depends on substrate availability of uridine-diphosphate-N-acetyl-glucosamine, UDP-GlcNAc [38,39]. This substrate is the donor of GlcNAc for O-GlcNAcylation and is produced by the hexosamine biosynthetic pathway (HBP) [40] (Figure 1). This anabolic pathway shifts glucose metabolism to produce high energy sugar intermediates for different types of glycosylation. Accordingly, pharmacological or genetic decrease in HBP flux through inhibition of HBP rate-limiting enzyme glutamine-fructose amino transferase (GFAT, expressed in two isoforms, GFAT1 and GFAT2, encoded by the genes GFPT1 and GFPT2, respectively) significantly lowers O-GlcNAcylation in vitro and in vivo [41,42,43,44]. Increases in UDP-GlcNAc levels, i.e., bypassing GFAT activity by adding glucosamine, enhances O-GlcNAcylation in vitro and in vivo [45,46,47]. UDP-GlcNAc is considered a central metabolic sensor because it requires inputs from nucleotide, carbohydrate, fatty acid, and amino acid metabolism [39,48].

Indeed, O-GlcNAcylation regulates several cell processes, including proliferation [49], differentiation [50], motility [51], and gene transcription [52]. This task is performed by the direct effects of O-GlcNAcylation [30,31], by noncatalytic functions of OGT [53], or by intricate crosstalk with other PTMs, such as phosphorylation [54,55] and ubiquitination [56]. At this point, an important gap is still being discussed: How is O-GlcNAcylation so abundant and diverse (i.e., it has been found in over 5000 proteins to be modified [57,58]) when cells have only one transferase and one glycosidase? Interactions between OGT or OGA with binding partners could promote selective O-GlcNAcylation of target proteins in a specific cell context [59,60,61]. How this is performed, however, requires further examination. 

O-GlcNAc homeostasis operates under a physiological threshold of changes in cellular O-GlcNAcylation levels [30]. Changes beyond this threshold for excessive periods of time, however, would generate pathophysiological processes. This model fits with data observed in diabetes [62,63,64], cancer [65,66,67], and cardiovascular [46,68,69,70] and neurodegenerative diseases [71,72,73,74]. In diabetes, the hyperglycemic milieu has been shown to shift cell metabolism towards increased HBP, increasing UDP-GlcNAc production and enhancing protein O-GlcNAcylation [75,76]. This mechanism mediates hyperglycemia-induced cytotoxicity [76]. In cancer cells, increased proliferation is supported by metabolic adaptions such as enhanced glucose [77] and glutamine [78] uptake and enhanced ATP production through glycolysis, a condition termed Warburg effect [79]. Importantly, HBP-mediated UDP-GlcNAc production sustains tumor development through aberrant glycosylation and protein O-GlcNAcylation [80,81]. Acute increases in O-GlcNAcylation protect the brain against stroke [71] and ameliorate cognitive decline in elder mice [74]. This protective effect of acute elevations in O-GlcNAcylation is also seen in trauma-induced cardiac dysfunction [46]. Chronic changes in O-GlcNAcylation levels, however, are associated with cardiovascular diseases, such as diabetes-induced arrhythmia [70] and heart failure [82], and the development of neurodegenerative diseases, including Alzheimer’s [73]. This model also applies to renal physiology and pathophysiology, as discussed below.

## 3. O-GlcNAcylation in Renal Physiology

Transcriptomic and proteomic analysis of different nephron segments have demonstrated that OGT, OGA, and GFAT expression are higher at medullary segments, including the thin descending limb (TDL), thin and thick ascending limb of the loop of Henle, and collecting duct [83,84]. Cortical segments, including proximal tubules and distal tubules, have lower but significant expression levels. Advances in tissue-specific gene editing tools have been used to study the importance of O-GlcNAcylation in specific extrarenal cells [85,86,87,88,89,90]. Ono et al. (2017) [91] accessed the importance of O-GlcNAcylation for podocytes, differentiated epithelial cells involved in selective permeability through the glomerular filtration barrier [92]. Congenital deletion of OGT in podocytes caused progressive podocyte loss, albuminuria, glomerulosclerosis, and subsequent tubule-interstitial injury. This was associated with reduced podocin expression, podocyte effacement, and disruption of slit diaphragm structure. Interestingly, tamoxifen-induced OGT deletion in adult mice did not cause significant changes in albuminuria or podocyte structure [91]. Thus, OGT and, consequently, O-GlcNAcylation appear to be essential for podocyte development, but not necessary for differentiated podocyte function in physiological conditions.

Sugahara and colleagues (2019) [93], using a tamoxifen-induced, proximal tubule-specific OGT deletion in adult mice, did not observe changes in renal function or urine excretion of different solutes reabsorbed by PTECs (proximal tubular epithelial cells) under ad libitum-fed conditions. However, in fasting conditions, OGT deletion caused PTEC apoptosis associated with widespread reduction in reabsorption capacity, reflected by urinary loss of glucose, albumin, amino acids, phosphate, calcium, and other solutes [93]. This was not associated with significant changes in the mRNA expression of different transporters such as SGLT2 and NaPiIIa, involved in the reabsorption of glucose and phosphate, respectively. Rather, significant metabolic changes were observed. Absence of OGT decreased PTEC lipolysis, the principal energy source of PTECs in fasting conditions. Furthermore, OGT KO exacerbated tubule interstitial injury induced by a high fat diet [93]. These results are in line with previous reports that changes in PTECs metabolism is associated with renal damage [94].

So far, the results available in the literature point to a role for OGT in early glomerulus development and maintenance of physiological proximal tubule metabolism, contributing to proper reabsorption capacity in this segment [91,93]. Future directions should include the use of OGA deletion models for understanding the impact of increased O-GlcNAcylation for renal physiology, as well as assessing the influence of O-GlcNAcylation in different renal cell types.

## 4. O-GlcNAcylation in Renal Pathophysiology

### 4.1. Kidney Diseases Classification

According to Kidney Disease Improving Global Outcomes (KDIGO) guidelines, kidney disease can be classified as: (1) acute kidney injury (AKI), consisting of estimated glomerular flow rate (eGFR) <60 mL/min/1.73 m^2^ for up to 7 days; (2) acute kidney disease (AKD), characterized by sustained decreased eGFR for 7 days up to 3 months; (3) chronic kidney disease (CKD), characterized by decreased eGFR and significant renal damage for more than 3 months [95]. Each classification can also be stratified by the degree of decrease in eGFR or the amount of urinary albumin excretion (albuminuria) [96]. A new condition, characterized by minimal change in glomerular structure and function in the presence of urinary tubular injury biomarkers, such as microalbuminuria, β2-microglobulin, and urinary neutrophil gelatinase-associated lipocalin (NGAL), KIM-1 and γ-GT [97,98]. This condition, so called subclinical acute kidney injury (subAKI), is associated with a higher risk of kidney disease progression and higher mortality [98,99,100,101], representing a window for early treatment of patients at risk for developing CKD.

Kidney diseases are associated with environmental factors, such as exposure to nephrotoxic xenobiotics (i.e., gentamicin [102], cisplatin [103]); genetic factors, such as mutations in essential genes for renal physiology (i.e., Collagen IV gene ColIV, a cause for Al-port Syndrome [104]); and acquired factors, including chronic degenerative comorbidities such as essential hypertension [105,106], diabetes [107,108], and obesity [109]. Indeed, hypertensive nephropathy and diabetic kidney disease represent the mechanism involved in the development of 75% of all CKD cases diagnosed [106,107]. 

Importantly, AKI and CKD are not mutually exclusive forms of kidney diseases. A patient with CKD can experience insults of AKI, which is associated with CKD progression [110]. Conversely, patients that have developed AKI are at higher risk of developing CKD in the long term [111,112]. This complex relationship between AKI and CKD is being investigated, but a common factor involved in poor prognosis for both is the degree of albuminuria. The rate of urinary albumin excretion is a risk factor for all-cause mortality [113,114] and predicts the progression of end-stage renal disease in hypertensive and diabetic patients [115,116]. Corroborating an active role of albuminuria in the progression of renal diseases, antiproteinuric treatments significantly improve patient’s survival and lower renal function decline of patients with established CKD [117,118].

### 4.2. Short Sweetness Seems Good–Renal Protective Effects of Acute Elevations in O-GlcNAcylation

Acute increases in O-GlcNAcylation attenuate acute injuries in several organs, including the heart [46] and the brain [74]. The outcome of this strategy in AKI severity and prognosis has been also investigated. Glucosamine treatment, a strategy that bypasses GFAT regulation and directly raises UDP-GlcNAc production [119], protected different organs, including the kidneys, against hemorrhage-induced damage [120]. To note, glucosamine effects could be either induced through O-GlcNAcylation-dependent or -independent mechanisms [i.e., through modulation of extracellular N-glycosylation [121]). However, acute treatment with the OGA inhibitor (PUGNAc) also attenuated hemorrhage-induced renal damage, suggesting that this mechanism involves protein O-GlcNAcylation [122]. In mice models of ischemia/reperfusion injury (IRI), there is a reduction in renal O-GlcNAcylation and OGT expression, and increased OGA expression [123]. Acute intravenous injection of glucosamine ameliorated IRI-induced glomerular dysfunction, measured by plasma creatinine and blood urea nitrogen [123].

AKI is also a result of exposure to nephrotoxic agents. Several medications are known for inducing AKI, including contrast for image exams, several antibiotics and chemotherapy, such as gentamycin [102] and cisplatin [103], respectively. Increasing O-GlcNAcylation through glucosamine alleviates PTEC apoptosis and oxidative stress in a contrast-induced AKI mice model [124]. The protective effect of remote ischemic preconditioning on contrast-induced AKI requires an acute elevation in O-GlcNAcylation [125].

What is the mechanism behind the protective effects of acute increases in protein O-GlcNAcylation? O-GlcNAcylation is a cell-intrinsic, stress-responsive mechanism [126]. Biochemical studies show that elevated O-GlcNAcylation levels regulate multiple pathways involved in stress response, such as heat shock protein (HSP), chaperones and multiple proteins from the ER stress response [126]. OGT is required for thermotolerance through an HSP-dependent and -independent manner [127]. Interestingly, increased HSPB1 and HSP 70 expression protects against AKI induced by oxidative stress and ischemia/reperfusion injury [128,129]. O-GlcNAcylation modulates the stability of hypoxia-inducible factor 1α (HIF1α) [130], a protective factor against I/R-induced renal damage [131]. O-GlcNAcylation is required for the formation of stress granules, an intracellular organelle involved in the clearance of misfolded proteins during ER stress [132,133]. Indeed, stress granules protect proximal tubules from apoptosis induced by acute toxicants and ischemic injury in vitro [134].

Thus, the current data suggests a protective role of increased O-GlcNAcylation in acute kidney damage conditions. Further studies should clarify the molecular mechanisms involved, despite a possible explanation through regulation of different stress response pathways.

### 4.3. Too Sweet to Handle–Prolonged Changes in O-GlcNAcylation Are Involved in Development of Kidney Diseases

Chronic changes in protein O-GlcNAcylation homeostasis are associated with the development of chronic degenerative diseases, such as Alzheimer’s [73], obesity [135], hypertension [69], and diabetes [62]. Regarding diabetes, hyper-O-GlcNAcylation has been proposed as a central mechanism for diabetes-induced organ damage [62,64,136,137]. Hyperglycemia is associated with increased O-GlcNAcylation levels in the glomerular and tubular cells in diabetic patients [138]. In vitro, high glucose promotes plasminogen activator inhibitor-1 (PAI-1), fibronectin, and transforming growth factor-β (TGF-β) expression in an O-GlcNAc-dependent manner in mesangial cells [139]. The mechanism involves p38 MAPK activation [139] and O-GlcNAcylation of carbohydrate response element-binding protein (ChREBP) [140]. These mechanisms could partially explain the progression of glomerular fibrosis and mesangial matrix deposition in diabetic conditions, a factor directly correlated with renal function decline [141,142]. Glomerular fibrosis increases albumin permeability through the glomerular filtration barrier [143] promoting the “albumin overload” condition in proximal tubule cells. Albumin overload promotes tubule-interstitial injury and tubular apoptosis through different mechanisms [144], including lysosomal permeability [145], induction of pro-apoptotic signaling [146], and induction of ER stress [147].

In addition, high glucose-induced O-GlcNAcylation inhibits megalin-mediated albumin endocytosis [148], the final gatekeeper mechanism against proteinuria and albuminuria [4]. Indeed, early albuminuria in diabetic mice is associated with impaired proximal tubule reabsorption [149]. The mechanism involves the inhibitory O-GlcNAcylation of AKT, inhibiting its activity [148]. AKT was previously shown to regulate albumin endocytosis in several points, such as promoting megalin mRNA synthesis [146] and megalin trafficking through endolysosomes [14,150]. AKT is a target of O-GlcNAcylation [151,152,153].

The possible role of increased O-GlcNAcylation in the development of proteinuria was addressed in animal models. Hodrea and colleagues [154] demonstrated that the treatment with SGLT2 inhibitor dapaglifozin significantly ameliorates DKD progression in a model of streptozocin-induced diabetes. Dapaglifozin reduced albuminuria, ameliorated plasma creatinine and BUN (glomerular injury markers), decreased KIM-1 and NGAL excretion (tubular injury markers), and reduced renal fibrosis and tubule-interstitial injury. Renal protective effects of dapaglifozin were associated with reduced proximal tubule O-GlcNAcylation [154]. These results corroborate clinical evidence that SGLT2 inhibitors ameliorate renal and cardiovascular function in diabetic patients [155] and suggest that reducing renal O-GlcNAcylation might be beneficial to halt diabetic kidney disease progression.

Interestingly, increased O-GlcNAcylation has been observed in cardiac and vascular tissues in hypertensive patients and animal models [69,156]. This was also observed in the renal cortex of spontaneously hypertensive rats (SHR) [157]. Hyper-O-GlcNAcylation was shown to inhibit proximal tubule albumin reabsorption through decreasing surface megalin expression due to its internalization. Accordingly, decreasing pharmacological inhibition of GFAT reduced albuminuria and proteinuria in a pressure-independent manner. The proposed mechanism could be mediated by megalin O-GlcNAcylation [157].

Progression of autosomal dominant polycystic kidney disease, ADPKD, an autosomal syndrome induced by mutations in intraflagellar transport (IFT) proteins, has also been associated with increased renal O-GlcNAcylation [158]. Renal inflammation, cystogenesis, and histological damage was correlated with renal O-GlcNAcylation levels, indicating that chronically elevated O-GlcNAcylation could be a hallmark of kidney diseases from different etiologies, ranging from acquired (diabetic and hypertensive) to genetic (ADPKD).

## 5. Conclusions and Perspectives

Unbalanced O-GlcNAcylation has been identified and discussed in several chronic degenerative diseases such as Alzheimer’s, diabetes, and cardiovascular diseases. Here, we discussed evidence that renal function is also a subject of O-GlcNAc homeostasis. In accordance with other tissues, acute modifications in O-GlcNAcylation appear to induce a protective effect in renal function, while chronic dysregulation is involved in the development and progression of renal diseases (Figure 2). Despite clinical trials of molecules targeting O-GlcNAc homeostasis being underway (MK-8719 from Merck and LY-3372689 from Eli Lilly as potential OGA inhibitors), approaches for targeted inhibitor delivery should be considered to avoid off-target effects. Furthermore, fundamental questions are still unanswered: (1) How O-GlcNAcylation is tightly regulated while maintaining great diversity [56]? (2) How O-GlcNAcylation correlates with other PTMs? Even though an important crosstalk between O-GlcNAcylation and phosphorylation is well-established, the interaction of O-GlcNAcylation with other PTMs such as acetylation, methylation, and others is still poorly understood. (3) How is O-GlcNAcylation regulated in different tissues in physiological and pathophysiological contexts? These questions require advances in detection methods for O-GlcNAcylated sites and cell-specific knock-out models that are being currently developed by different groups [159,160,161]. Current detection limitations still delay the development of the protein O-GlcNAcylation field. Compared to protein phosphorylation, the development of specific antibodies to detect single site O-GlcNAcylation have intrinsic biological limitations (detection of GlcNAc-modified sites by antibodies). Additionally, the detection tools currently being employed are complex and expensive, such as O-GlcNAc labelling followed by mass spectrometry analysis [159]. Recent advances such as chemical reporters and biorthogonal reactions, however, are promising strategies to foster future protein O-GlcNAcylation studies [162].

Defining the role of protein O-GlcNAcylation is an interesting future perspective for the renal physiology field given the great renal cellular diversity (approximately 73 cell types contained within an adult kidney [163]). Understanding the importance of this mechanism in a cell-dependent manner should significantly advance the comprehension of how renal function maintains homeostasis in challenging conditions. Furthermore, it should enlighten possible therapeutic opportunities to halt the development and progression of renal diseases, positively impacting global healthcare programs worldwide.

## Figures and Tables

**Figure 1 ijms-23-11260-f001:**
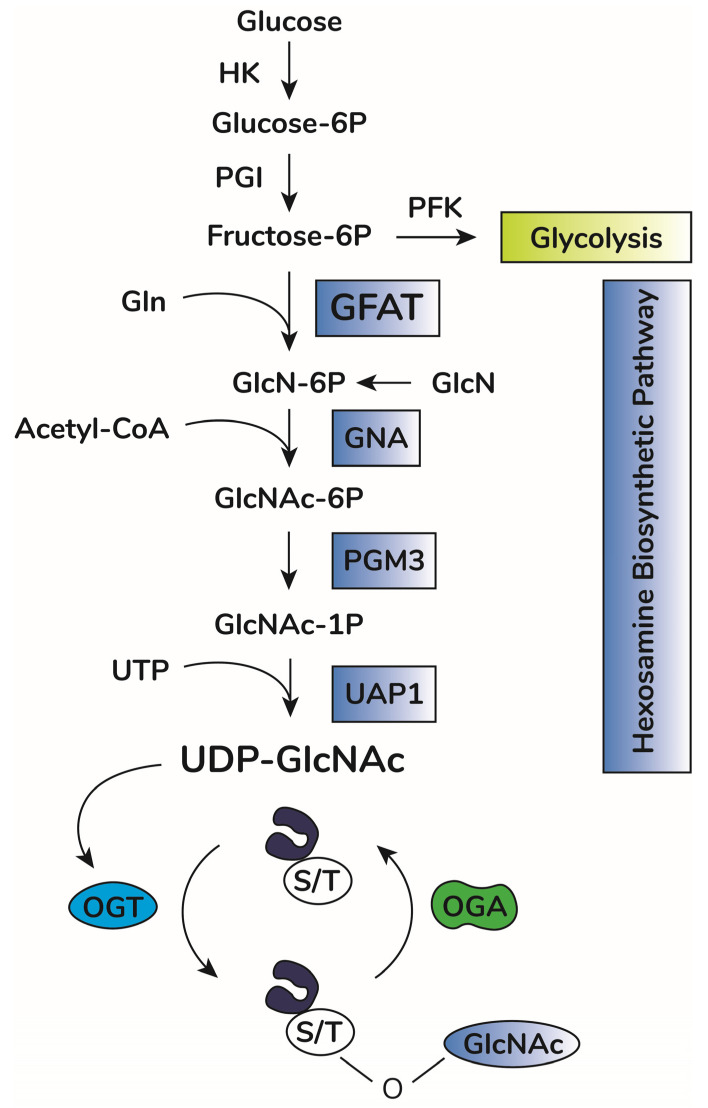
O-GlcNAc cycling scheme. Upon entry in cells, glucose is rapidly phosphorylated to glucose-6-phosphate (Glucose-6P) by hexokinase (HK). Glucose-6P is isomerized by phosphoglucose isomerase (PGI), producing fructose-6-phosphate (Fructose-6P), a substrate for both phosphofructokinase (PFK) of the glycolytic pathway or glucosamine-fructose amino transferase (GFAT), the rate-limiting reaction of the hexosamine biosynthetic pathway (HBP). GFAT requires glutamine (Gln) as the amine donor for generating glucosamine-6-phosphate (GlcN-6P), which is then N-acetylated by glucosamine-6-phosphate N-acetyltransferase (GNA1), producing N-acetyl-glucosamine-6-phosphate (GlcNAc-6P). This step requires acetyl-CoA as the acetyl donor. GlcNAc-6P is transformed in GlcNAc-1P by phosphoacetylglucosamine mutase (PGM3). Using UTP as the nucleotide donor, UDP-N-acetylglucosamine pyrophosphorylase (UAP1) produces uridine-diphosphate N-acetyl glucosamine (UDP-GlcNAc). This molecule is the substrate of O-GlcNAc transferase (OGT) for protein O-GlcNAcylation by adding O-linked GlcNAc moieties at serine or threonine residues of target proteins. O-GlcNAcase (OGA) removes GlcNAc residues, counteracting OGT activity.

**Figure 2 ijms-23-11260-f002:**
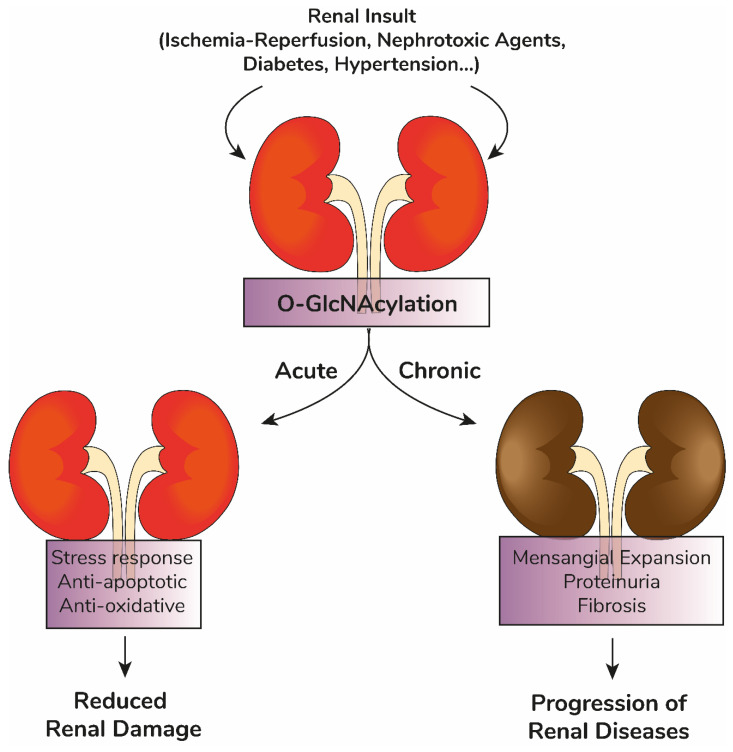
Proposed model. The kidneys are possible targets of acute (ischemia-reperfusion injury, nephrotoxic agents) or chronic (diabetes, hypertension) insults. Renal O-GlcNacylation is associated with different outcomes in these conditions: acute increases in O-GlcNAcylation are associated with stress response, decreased apoptosis, and oxidative stress, reducing renal damage; chronic increases in renal O-GlcNAcylation observed in diabetes and hypertension are associated with mesangial expansion, proteinuria, and renal fibrosis, inducing the progression of renal diseases.

## Data Availability

Not applicable.

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
