# Peer review of "O-GlcNAcylation in Renal (Patho)Physiology"

_ijms, 2022, doi:10.3390/ijms231911260_

Round 1
Reviewer 1 Report
In this paper, author discuss the relationship between O‑GlcNAcylation and kidney disease, in a short-term, elevated O‑GlcNAcylation have a Renal protective function, while long-term high level of O‑GlcNAcylation promote the development of kidney diseases. The most important problem of paper is it seems that author lack much relevant knowledge of O‑GlcNAc, which limit the quality of this paper. And author is also very careless, I can find many unnecessary mistakes.
1. Line 23-30, please delete this meaningless paragraph.
2. Line 61, I think just list several prevalent PTMs is enough, “and others” is not necessary.
3. Line 63, “and” should be replaced by “or”, “in” should be replaced by “on”.
4. Figure 1, this figure is too simple and careless. First, author should mark that O-GlcNAc just link to serine or threonine residue the on protein. Second, author should draw main metabolites in HBP, especially the rate-limited step catalyzed by GFAT, which use glutamine to convert fructose-6-phosphate and glucosamine-6-phosphate.
5. In the section of “2. O‑GlcNAcylation”, author should also discuss relationship between cancer and diabetes, which are much more important than cardiovascular diseases and neurodegenerative diseases. O‑GlcNAc is a metabolic sensor, which derives from glucose metabolism. It is a well-known PTM mainly depend on it is involved in metabolic dieses, cancer and diabetes.
6. I don’t understand, there are two section#3” O‑GlcNAcylation in Renal Physiology”, I think in first section#3, author want to discuss how those enzymes OGT, OGA and GAFT involved in kidney disease. It is very chaos.
7. In the section of “4. Conclusion and Perspectives”, author should discuss that O-GlcNAc research is still limited worldwide, one of the main problems is that lack of reliable detection tools, such as specific antibody. It is a very big problem that hamper development of O-GlcNAc research.
Author Response
We appreciate the constructive concerns raised by the reviewers. All questions were promptly answered as indicated below. The new texts added were marked in bold. We corrected the final Reference list to include the new references added.
Reviewer #1:
In this paper, author discuss the relationship between O-GlcNAcylation and kidney disease, in a short-term, elevated O-GlcNAcylation have a Renal protective function, while long-term high level of O-GlcNAcylation promote the development of kidney diseases. The most important problem of paper is it seems that author lack much relevant knowledge of O-GlcNAc, which limit the quality of this paper. And author is also very careless, I can find many unnecessary mistakes.
1) Line 23-30, please delete this meaningless paragraph.
Answer: We removed the paragraph as suggested.
2) Line 61, I think just list several prevalent PTMs is enough, “and others” is not necessary.
Answer: We removed the sentence “and others” in the final version of the manuscript.
3) Line 63, “and” should be replaced by “or”, “in” should be replaced by “on”.
Answer: We made de corrections according to the reviewer suggestion.
4) Figure 1, this figure is too simple and careless. First, author should mark that O-GlcNAc just link to serine or threonine residue the on protein. Second, author should draw main metabolites in HBP, especially the rate-limited step catalyzed by GFAT, which use glutamine to convert fructose-6-phosphate and glucosamine-6-phosphate.
Answer: We appreciate the reviewer concern and have improved the Figure 1 scheme. We have added information about the target residues for protein O-GlcNAcylation, and detailed the hexosamine biosynthetic pathway, adding intermediates and nutrients involved in UDP-GlcNAc production. We also changed Figure 1 legend from:
Page 3, line 90:
“Figure 1. O-GlcNAc cycling scheme. Hexosamine biosynthetic pathway receives the input of aminoacids, glucose, nucleotide, and fatty acids to produce uridine-diphosphate N-Acetyl glucosamine (UDP-GlcNAc). This molecule is the substrate of O-GlcNAc transfer-ase (OGT) for protein O-GlcNAcylation. O-GlcNAcase (OGA) removes GlcNAc residues, counteracting OGT activity.”
To:
“Figure 1. O-GlcNAc cycling scheme. Upon entry in cells, glucose is rapidly phosphorylated to glucose-6-phosphate (Glucose-6P) by hexokinase (HK). Glucose-6P is isomerized by phosphoglucose isomerase (PGI), producing fructose-6-phosphate (Fructose-6P), a substrate for both phosphofructokinase (PFK) of the glycolytic pathway or glucosamine-fructose aminotransferase (GFAT), the rate-limiting reaction of the hexosamine biosynthetic pathway (HBP). GFAT requires glutamine (Gln) as the amine donor for generating glucosamine-6-phosphate (GlcN-6P), which is then N-acetylated by glucosamine-6-phosphate N-acetyltransferase (GNA1), producing N-acetyl-glucosamine-6-phosphate (GlcNAc-6P). This step requires acetyl-CoA as the acetyl donor. GlcNAc-6P is transformed in GlcNAc-1P by phosphoacetylglucosamine mutase (PGM3). Using UTP as the nucleotide donor, UDP-N-acetylglucosamine pyrophosphorylase (UAP1) produces uridine-diphosphate N-Acetyl glucosamine (UDP-GlcNAc). This molecule is the substrate of O-GlcNAc transferase (OGT) for protein O-GlcNAcylation by adding O-linked GlcNAc moietes at serine or threonine residues of target proteins. O-GlcNAcase (OGA) removes GlcNAc residues, counteracting OGT activity.”
5) In the section of “2. O-GlcNAcylation”, author should also discuss relationship between cancer and diabetes, which are much more important than cardiovascular diseases and neurodegenerative diseases. O-GlcNAc is a metabolic sensor, which derives from glucose metabolism. It is a well-known PTM mainly depend on it is involved in metabolic dieses, cancer and diabetes.
Answer: We agree with the reviewer. To address this concern, we have added the following paragraph:
Page 4, line 117:
“ …This model fits with data observed in diabetes [60-62], cancer [63-65], cardiovascular [66-69] and neurodegenerative diseases [70-73]. In diabetes, the hyperglycemic milieu has been shown to shift cell metabolism towards increased HBP, increasing UDP-GlcNAc production and enhancing protein O-GlcNAcylation [74,75]. This mechanism mediates hyperglycemia-induced cytotoxicity [75]. In cancer cells, increased prolifera-tion is supported by metabolic adaptions such as enhanced glucose [76] and glutamine [77] uptake and enhanced ATP production through glycolysis, a condition termed War-burg effect [78]. Importantly, HBP-mediated UDP-GlcNAc production sustains tumor development through aberrant glycosylation and protein O-GlcNAcylation [79,80]….
6) I don’t understand, there are two section#3” O-GlcNAcylation in Renal Physiology”, I think in first section#3, author want to discuss how those enzymes OGT, OGA and GAFT involved in kidney disease. It is very chaos.
Answer: We appreciate the reviewer commentary. The first section #3 compiles data about the role of protein O-GlcNAcylation in renal physiology. The second section #3 is section #4, where we discuss kidney disease and the role of protein O-GlcNAcylation in this process. We corrected this issue in the corrected version of the manuscript.
7) In the section of “4. Conclusion and Perspectives”, author should discuss that O-GlcNAc research is still limited worldwide, one of the main problems is that lack of reliable detection tools, such as specific antibody. It is a very big problem that hamper development of O-GlcNAc research.
Answer: We agree with the reviewer suggestion. To clarify this issue, we added the following text:
Page 7, line 312:
“Current detection limitations still delay the development of protein O-GlcNAcylation field. Compared to protein phosphorylation, the development of specific antibodies to detect single site O-GlcNAcylation have intrinsic biological limitations (detection of GlcNAc-modified sites by antibodies). Additionally, detection tools currently being em-ployed are complex and expensive, such as O-GlcNAc labelling followed by mass spec-trometry analysis [158]. Recent advances such as chemical reporters and biorthogonal reactions, however, are promising strategies to foster future protein O-GlcNAcylation studies [161].”

Reviewer 2 Report
This is an interesting review which raises an important question about the O-GlcNAc homestasis balance in the context of kidney functions and diseases. The authors provide information about contrasting outcomes of acute and chronic elevation of O-GlcNAc in kidney and propose a respective model. Overall this model deserves attention and suggests a complex regulatory functions of O-GlcNAcylation in health and disease. I noticed only several issues which need few corrections or clarifications:
Line 32 and through the whole text: aminoacids should be amino acids (add space).
Line 72: there is no gene symbol OGT1; it might be OGT.
Line 158: should be Pathophysiology.
Line 234: do not mention Alzheimer in the context of this statement, because Alzheimer disease is associated with a decrease in O-GlcNAcylation (the reference 102 reports that the O-GlcNAcylation level in AD brain extracts was decreased as compared to that in controls). The authors may consider to mention that the balance of O-GlcNAc homeostasis (increase or decrease) might be essential in the etiologies of chronic diseases (e.g. diabetes and cancer versus Alzheimer and Parkinson) (PMID: 30626734).
Author Response
This is an interesting review which raises an important question about the O-GlcNAc homestasis balance in the context of kidney functions and diseases. The authors provide information about contrasting outcomes of acute and chronic elevation of O-GlcNAc in kidney and propose a respective model. Overall this model deserves attention and suggests a complex regulatory functions of O-GlcNAcylation in health and disease. I noticed only several issues which need few corrections or clarifications:
We appreciate the constructive concerns raised by the reviewer. All questions were promptly answered as indicated below. The new texts added were marked in bold. We corrected the final Reference list to include the new references added.
1) Line 32 and through the whole text: aminoacids should be amino acids (add space).
Answer: We appreciate the reviewer concern and have corrected this issue in the corrected version of the manuscript.
2) Line 72: there is no gene symbol OGT1; it might be OGT.
Answer: We appreciate the reviewer concern and have corrected this issue in the corrected version of the manuscript.
3) Line 158: should be Pathophysiology.
Answer: We appreciate the reviewer concern and have corrected this issue in the corrected version of the manuscript.
4) Line 234: do not mention Alzheimer in the context of this statement, because Alzheimer disease is associated with a decrease in O-GlcNAcylation (the reference 102 reports that the O-GlcNAcylation level in AD brain extracts was decreased as compared to that in controls). The authors may consider to mention that the balance of O-GlcNAc homeostasis (increase or decrease) might be essential in the etiologies of chronic diseases (e.g. diabetes and cancer versus Alzheimer and Parkinson) (PMID: 30626734).
Answer: We appreciate the concern. To clarify this issue, we changed the sentence:
Page 6, line 245:
“Chronic elevations of O-GlcNAcylation are associated with the development of chronic degenerative diseases, such as Alzheimer [102], obesity [121], hypertension [97] and diabetes [122].”
To:
“Chronic changes in protein O-GlcNAcylation homeostasis are associated with the development of chronic degenerative diseases, such as Alzheimer [72], obesity [134], hypertension [67] and diabetes [60].”

Reviewer 3 Report
Dias and the co-authors discussed the impact of protein O-GlcNAcylation on physiological renal function, disease conditions. They have also discussed the future possibilities of this field. I suggest accepting the manuscript as it is. Please include the following references.
1. O-GlcNAc: Epigenetic modifier linked to X-linked intellectual disability. Konzman, D. et al. Front. Genet., 2020, 11, 605263. doi: 10.3389/fgene.2020.605263
2. Olivier-Van Stichelen, S. et. al. Find out if your protein is O-GlcNAc modified: The O-GlcNAc database. The FASEB Journal 36 (2022)
Author Response
We appreciate the constructive concerns raised by the reviewer. All suggestions were promptly answered as indicated below. The new texts added were marked in bold. We corrected the final Reference list to include the new references added.
Dias and the co-authors discussed the impact of protein O-GlcNAcylation on physiological renal function, disease conditions. They have also discussed the future possibilities of this field. I suggest accepting the manuscript as it is.
1) Please include the following references.
- O-GlcNAc: Epigenetic modifier linked to X-linked intellectual disability. Konzman, D. et al. Front. Genet., 2020, 11, 605263. doi: 10.3389/fgene.2020.605263
- Olivier-Van Stichelen, S. et. al. Find out if your protein is O-GlcNAc modified: The O-GlcNAc database. The FASEB Journal 36 (2022)
Answer: We appreciate the reviewer comments and added the suggested references within the manuscript.

Round 2
Reviewer 1 Report
The authors have modified this paper and addressed my questions properly, I have no further suggestion.